# Health Outcomes Associated with Olive Oil Intake: An Umbrella Review of Meta-Analyses

**DOI:** 10.3390/foods13162619

**Published:** 2024-08-21

**Authors:** Manuela Chiavarini, Patrizia Rosignoli, Irene Giacchetta, Roberto Fabiani

**Affiliations:** 1Department of Biomedical Sciences and Public Health, Section of Hygiene, Preventive Medicine and Public Health, Polytechnic University of the Marche Region, 60126 Ancona, Italy; m.chiavarini@staff.univpm.it; 2Department of Chemistry, Biology and Biotechnology, University of Perugia, 06123 Perugia, Italy; patrizia.rosignoli@unipg.it; 3Local Health Unit of Bologna, Department of Hospital Network, Hospital Management of Maggiore and Bellaria, 40124 Bologna, Italy

**Keywords:** olive oil consumption, chronic diseases, non-communicable diseases, review/umbrella review, health outcomes

## Abstract

Several studies suggested a negative association between olive oil (OO) consumption and the risk of several chronic diseases. However, an attempt to systematically search, organize, and evaluate the existing evidence on all health outcomes associated with OO consumption is lacking. The objective of this review is to describe the multiple health outcomes associated with OO consumption. The Medline, Scopus, and Web of Science databases were searched through 5 April 2024. The selected studies met all of the following criteria: (1) a meta-analysis of both observational (case–control and cohort studies) and interventional studies (trials), (2) an evaluation of the association between OO consumption, mortality, and/or the incidence of non-communicable/chronic degenerative diseases, and (3) a study population ≥18 years old. Two independent reviewers extracted the relevant data and assessed the risk of bias of individual studies. The PRISMA statement and guidelines for the Integration of Evidence from Multiple Meta-Analyses were followed. The literature search identified 723 articles. After selection, 31 articles were included in this umbrella review. The primary health benefits of OO were observed in cardiovascular diseases and risk factors, cancer, mortality, diabetes, and specific biomarkers related to anthropometric status and inflammation. As a key component of the Mediterranean diet, OO can be considered a healthy dietary choice for improving positive health outcomes.

## 1. Introduction

Olive oil is obtained from the fruit of the olive tree (*Olea europaea* L.), exclusively by mechanical and physical means under certain conditions, particularly thermal conditions, that do not result in alterations in the oil, which has not undergone any treatment other than washing, decantation, centrifugation, and filtration [1]. Olive oil is an important element of the Mediterranean diet, distinguished as a Cultural Heritage of Humanity by UNESCO in 2010 [2]; in fact, olive oil represents the main source of meal fat [3].

Epidemiologic evidence has consistently shown that an increased consumption of olive oil is associated with a reduced risk of various chronic diseases. The findings from several key studies illustrate the broad health benefits of olive oil, highlighted particularly in relation to cardiovascular disease, cancer, type 2 diabetes, body composition, blood pressure, inflammation, endothelial function, and hemostasis [4,5,6,7,8,9,10]. The largest trial on the Mediterranean diet tested supplementation with extra virgin olive oil in one arm and reported a significant reduction in hard endpoint compared to the control group [11].

Olive oil is rich in monounsaturated fatty acids (oleic acid) (MUFAs) [12], unsaponifiable compounds (phytosterols, triterpenes, squalene, pigments, etc.) [13], and hydrophilic compounds (polyphenols, tocopherol, etc.) [14]. MUFAs and polyphenols (such as oleuropein, hydroxytyrosol, and tyrosol) are important components that explain the protective role of olive oil in disease development [15,16,17]. Among the phenolic components of olive oil, oleuropein (OLP) is considered the most effective biomolecule [18,19].

Regarding the biochemical mechanisms through which olive oil components may exert their effects, some in vitro and animal studies have described the action on inflammation mediators, lipoprotein metabolism, endothelial function, and cell cycle regulation and metabolism [20,21,22,23,24,25,26].

Despite the nutritional and epidemiological studies cited, many questions about the role of olive oil in disease remain unanswered; moreover, causality is difficult to prove. Previous efforts to systematically appraise the evidence on olive oil have focused on single disease endpoints (e.g., CVD). Instead, in this review, we used the umbrella review methodology (i.e., the syntheses of existing systematic reviews with meta-analyses) [27,28,29] to capture the full spectrum of outcomes associated with olive oil intake and to systematically assess the quality and strength of the evidence across all health outcomes and medical conditions to highlight those with the strongest evidence.

## 2. Materials and Methods

The protocol of the current umbrella review was registered in the PROSPERO International Prospective Register of Systematic Reviews database (ID number: 42023450410, www.crd.york.ac.uk/PROSPERO (accessed on 14 August 2023)). This study was carried out according to the recommendations of the “Preferred Reporting Items for Systematic reviews and Meta-Analyses” (PRISMA) checklist [27].

The aim of this general review was to provide health decision-makers with information on a broad topic involving multiple outcomes. The specific questions included the following: “Is olive oil consumption beneficial for population health? Is the existing evidence strong and valid enough to assess the association between olive oil intake and health outcomes in adults?”

### 2.1. Search Strategy and Data Sources

Two authors (RF and MC) independently conducted a systematic literature search until 5 April 2024, without restriction, through three electronic databases: PubMed, http://www.ncbi.nlm.nih.gov/pubmed/ (accessed on 5 April 2024); Web of Science, http://wokinfo.com/ (accessed on 5 April 2024); and Scopus, https://www.scopus.com/ (accessed on 5 April 2024) using the keywords “olive oil” AND “meta-analysis” in the title or abstract. Furthermore, the reference lists of the included articles were manually examined to find additional relevant publications. In addition to an electronic search, the authors reviewed the references quoted in the full-text articles to intercept further interesting articles. We included meta-analyses of observational and intervention studies investigating olive oil as a possible determinant for any health outcome.

### 2.2. Selection Process

The list of selected studies, the removal of duplicates, and the selection of studies of interest were managed with Zotero. Two authors (RF and MC) independently screened the titles and abstracts of all remaining articles before assessing the full texts. A third reviewer (PR) solved any disagreements. We used the PICOS framework (Population/Intervention/Comparison/Outcome/Study design) to guide our selection criteria. We included studies that met the following inclusion criteria: (a) an evaluation of the association between olive oil intake, mortality, and/or the incidence of non-communicable/chronic degenerative diseases; (b) adults aged 18 years or over in order to avoid the influence of exposure time on the outcome assessed; (c) a healthy or unhealthy population; (d) a reported summary of the estimated effect size, such as odds ratios (ORs), hazard ratios (HRs), or risk ratios (RRs), and their corresponding 95% confident intervals (CIs) for observational studies; and (e) reported effects such as weighted or standardized mean differences with confidence intervals (95%) for intervention trials. We excluded studies (a) published in a language other than English; (b) on animal models; (c) without a defined control group or a defined comparator. We also excluded meta-analysis with oleic acid or olive oil phenols as single components instead of olive oil.

### 2.3. Data Extraction

Using a previously defined form, the authors independently extracted data (RF and PR from observational studies, MC and IG from intervention studies). Any disagreements were resolved through discussion. From each meta-analysis, data extracted included the first author’s last name, year of publication, type of outcomes (cancer or non-cancer), number and design of included studies, comparison criteria, study population characteristics, number of events, sample size, type of reported effect size (e.g., relative risk/hazard ratio, odds ratio, mean difference), corresponding 95% CI, tools used to assess risk of bias and heterogenicity. Data were grouped according to the type of outcome.

### 2.4. Quality Assessment

The methodological quality of the included meta-analyses was independently assessed by two authors (RF and PR). In the event of a disagreement, a third investigator (either MC or IG) was involved. We evaluated the methodological quality of the included meta-analysis using the “A Measurement Tool to Assess systematic Reviews 2” (AMSTAR-2) questionnaire, which comprises 16 items. The overall assessment focused on deficiencies in the following seven critical areas: the protocol was defined before starting the study (study registration) (item 2); an exhaustive literature search was conducted (item 4); the rationale for the exclusion of individual studies was discussed (item 7); an appropriate method was used to analyze the risk of bias of individual studies (item 9); appropriate methods were used for the statistical combination of results (item 11); possible bias was considered in the discussion of results (item 13); the possible impact of publication bias was considered (item 15).

The overall confidence level ranged from high (no or one non-critical weakness), to moderate (more than one non-critical weakness), low (one critical weakness), and critically low (more than one critical weakness) [30].

### 2.5. Data Analysis 

For each meta-analysis, we presented the most adjusted estimated summary effect size with their 95% CI using random or fixed effects models. Dose–response analyses were also extracted from each study. Publication bias was assessed by the Egger regression asymmetry test [31]. The heterogeneity among studies was assessed by the I^2^ metric and Cochran’s Q test. For heterogeneity and publication bias, as also for other tests, *p* < 0.05 was adopted for the significance threshold because of the limited statistical power. 

## 3. Results

### 3.1. Study Selection

The systematic search yielded 723 records, 123 from PubMed, 392 from Scopus, and 208 from Web of Science. Before screening, 234 records were removed due to duplication. Of the 489 records screened for titles and abstract, 443 were excluded. Forty-six studies were assessed for eligibility, of which fifteen were excluded based on the full text [32,33,34,35,36,37,38,39,40,41,42,43,44,45,46]. The literature search process is depicted in the flow diagram shown in Figure 1.

Study characteristics are summarized in two tables, depending on the measure of association used: one for outcomes summarized as standardized mean difference (SMD), and another for outcomes summarized as Risk Ratio (RR) or Odds Ratio (OR). Eighteen articles used SMD [9,47,48,49,50,51,52,53,54,55,56,57,58,59,60,61], while fourteen articles employed RR [7,61,62,63,64,65,66,67,68,69,70,71,72]. Seven articles described the association between cancer and olive oil consumption [61,62,63,64,65,66], twenty articles investigated the association between cardiovascular risk or disease and olive oil [4,47,50,51,52,53,54,55,56,57,58,59,60,61,67,68,69,70], four articles the association between anthropometric indices and olive oil [47,48,49,54], three articles the association between inflammatory biomarkers and olive oil [9,14,47], two articles the association between diabetes and olive oil [4,7], one the association between pressure ulcers and olive oil [71], and five the association between mortality and olive oil [4,61,67,69,72].

### 3.2. Quality Assessment of the Meta-Analyses Included

Using the AMSTAR 2 questionnaire, we assessed the methodological quality of the meta-analyses included in this umbrella review. The quality judgement was rated high in four articles, moderate in six articles, low in two, and critically low in twenty. The main criticisms were often related to the absence of PICO components in the research question and inclusion criteria, inadequate search strategies for literature, an insufficient discussion of the potential impact of heterogenicity, and bias on the results. Additionally, the sources of funding for the studies included in the review were omitted. The overall judgment is reported in Table 1 and Table 2, whereas the item-by-item assessment for each included meta-analysis is detailed in Appendix A.

### 3.3. Cancers

Three articles addressed “All sites cancer” [61,62,65], six discussed “Breast cancer” [4,62,63,64,65,66], and two focused on digestive system cancers [62,65]. The minimum number of studies included was two [4], while the maximum was forty-five [62]. Most studies compared the highest versus quantities of olive oil consumption [61,62,63,64,65,66], while two evaluated increasing gram increment [4,61]. All studies included participants older than 18 years, with four studies considering both men and women [61,62,63,64,65,66], and three focusing solely on women [63,64,66]. Five outcomes were assessed using OR: “All sites” of Markellos 2022 [62], “Breast cancer” of Sealy 2021 [63], Xin 2015 [64], and all outcomes of Psaltopoulou 2011 [65], whereas the remaining studies used RR. All articles employed a random effect model. Among studies targeting “All sites”, two studies [62,65] concluded that olive oil is a protective factor against all type of cancers. Regarding “Breast cancer”, only four articles [62,64,65,66] identified a significant association between olive oil consumption and breast cancer. Conversely, olive oil consumption appears to protect against digestive cancer in all studies addressing the outcome [62,65], particularly the study of Markellos 2022, which detailed outcome in gastrointestinal, colorectal, and upper aerodigestive cancers. The quality of the studies was rated low for two articles [61,63], critically low for three [64,65,66], moderate for one [62], and high for one [6]. For further information, see Table 1.

### 3.4. Cardiovascular Disease and Risk Factors

#### 3.4.1. Association Measure RR or OR

Table 1 groups various cardiovascular disease outcomes, detailed as follows:Cardiovascular disease (CVD), assessed by six articles [4,61,67,68,69,70];Coronary heart disease (CHD), assessed by four articles [4,61,69,70];Stroke, assessed by four articles [4,61,69,70].

Studies included ranged from two [70] to nine [4]. The study population was always older than 18 years and included always men and women. The comparison was mainly obtained by increasing the quantity of olive oil [4,61,67,70], but also by comparison of highest versus lowest [61,67,69].

All studies assessing the association between CVD and olive oil consumption found a significant association [4,61,67,68,69,70]. Concerning CHD, all referenced articles reported a significant association [4,61,69,70] with olive oil consumption. The association between stroke and olive oil consumption was significant in all the articles that examined this outcome [4,61,69,70], except in the study by Ke 2024, which compared the highest versus the lowest consumption levels [61].

Quality of the studies was low for two articles [61,70], critically low for three articles [32,67,69], high for one [4].

#### 3.4.2. Association Measure SMD

Fifteen articles examined the association between cardiovascular risk factors using SMD [47,48,50,51,52,53,54,55,56,57,58,59,60]. The outcomes analyzed were (Table 2):Total Cholesterol (TC), considered in eight articles [47,48,50,53,56,60];Triglycerides (TG), considered in seven articles [47,48,53,57,58,59,60];HDL and LDL, considered in nine articles [47,48,50,53,56,57,59,60];VLDL, considered in one article [47];Ox LDL, considered in four articles [8,51,56,60];Apo A1 and Apo B, considered in two studies [47,59];Lp-A, considered in one article [47];HbA1c, considered in one article [7];“Metabolic syndrome”, considered in one article [54];Glycemic profile, considered in one article [54];Lipid profile, considered in one article [54];Glucose, considered in two articles [8,48];Fasting blood glucose (FBS), considered in two articles [7,55];Insulin level, considered in three articles [47,48,55];Homeostatic model assessment for insulin resistance (HOMA-IR), considered in two articles [47,55];Systolic Blood Pressure (SBP) and/or Diastolic Blood Pressure (DBP), considered in six articles [47,52,54,57,58,60];Malondialdehyde plasmatic (MDA), considered in three articles [51,57,60];Ferric-reducing ability of plasma, considered in one article [51],

The type of oil analyzed included olive oil in eleven studies [8,48,50,51,52,54,56,58,59,60], EVOO oil in two articles [47,55], and a mixture of oils in one article [54]. The comparison is low-fat diet and other oils [48], regular diet and other oils [47,53,54,58], LPOO [56,60], other oil type [7,8,59], olive oil [50,51,57], natural products or placebo [52], EVVO or OO or refined OO or Virgin OO [53]. The population ranges from 79 to 10.996 participants and it is composed in all of the studies by over-18 men and women, healthy and not.

The association between total cholesterol and olive oil was significant in only three studies [8,56,59]. A significant association is also observed between HDL/LDL and olive oil [8,50,51,56,59] and between MDA and olive oil [51,56]. Another significant association is observed between blood pressure and olive oil [52,57,58,60], and glycemic profile and olive oil [7,47].

The quality of the studies was rated as critically low in eleven studies [50,52,53,54,55,56,57,58,59,71], moderate in three [47,48,60], and high in one [7].

### 3.5. Pressure Ulcers

A single meta-analysis assessed the association between pressure ulcers and olive oil [71]. This high quality meta-analysis highlighted a reduced incidence of pressure ulcers following topical application of EVOO, with no adverse effect observed (Table 1).

### 3.6. Mortality

Five articles evaluated the association between all causes of mortality and olive oil [4,61,67,69,72], two addressed cardiovascular mortality and olive oil [61,69], and one discussed cancer mortality and olive oil [61] (Table 1). The number of studies included varied from four [72] to fourteen [61]. The population examined in all the studies consisted of mixed males and females over 18 years old [4,61,67,69,72].

Olive oil appears to have a significant protective effect on all-cause mortality in all but one study [72]. Similarly, a significant association between cardiovascular mortality and olive oil consumption was found in one study [61], but not in another [69]. No significant preventive effect was observed for cancer mortality and olive oil [4,61].

The quality was low for one study [61], critically low for three studies [67,69,72], and high for the study of Martinez-Gonzales 2022 [4].

For further information, see Table 1.

### 3.7. Antropometric Indices

Four meta-analysis [47,48,49,54] explored the association between olive oil and various anthropometric indices, including Body Mass Index (BMI) [47,48,49], Waist Circumference (WC) [47,48,49], Hip Circumference (Hip C) [49], WC/Hip C [47,49], Weight [47], Total Body Fat [49], muscle mass [49], and body composition [54]. The meta-analysis considered olive oil or EVOO or refined olive oil administered in various forms, either as a culinary ingredient (consumed after cooking or crude) or in capsules. Comparisons varied widely, ranging from a low-fat diet and other vegetable oils [48], to a regular diet with other vegetable oils or animal fats [47] and diets that were either standard, hypocaloric, or supplemented with nuts or PUFAs or a low-fat diet [49,54]. All of the studies were performed on a population over 18 years old, composed of men and women. One study enrolled only unhealthy subjects [48], and one both healthy and unhealthy subjects [54]. We found no significant differences in anthropometric indices between intervention and control groups. Only BMI was significantly lower in the population that consumed olive oil than in a population exposed to a low-fat diet and other oils [48].

The analysis of the quality of the studies shows moderate quality in three studies [47,48,49] and critically low quality in one [54].

### 3.8. Inflammatory Biomarkers

Three meta-analysis evaluated the relationship between olive oil consumption and inflammatory biomarkers [9,14,47], focusing on CRP [48,57,65], IL-6 [48,57,65], IL-10 [47], TNF-α [9,47], flow-mediated dilatation, Adiponectin, sE-Selectin, sP-Selectin, sICAM-1, and sVCAM-1 [9]. All three studies incorporated EVOO or olive oil as a culinary ingredient of the diet, and additionally, one study also explored the effects of olive oil administered in capsules [9]. Comparisons varied widely, ranging from a regular diet with other oils or butter [47], to a low-fat diet or a diet enriched with nuts and PUFAs, or other vegetable and animals oils, or a general healthy diet [9], to diets rich in saturated fatty acids or other oil [14]. The numbers of studies included in the metanalyses ranged from three to fourteen. The study populations were all over 18 years of age, and included both women and men, healthy and unhealthy in two studies [9,14,47]. A significant reduction in IL-6 was observed in the intervention groups in two of the three meta-analyses [9,14]. Similarly, a negative trend for CRP was noted among EVOO consumers, although this was significant in one study [9]. One article identifieda positive association for flow-mediated dilatation, adiponectin, TNF-α, sE-Selectin, sP-Selectin, sICAM-1, and sVCAM-1 [9], although significance was only achieved for flow-mediated dilatation and sE-Selectin. Two of the articles were rated as moderate in quality [14,47], while one was rated as critically low [9].

## 4. Discussion

Our findings suggest that olive oil intake generally plays a preventive role against cancer risk, particularly significant when analyses combine both case–control and cohort study designs, or when only case–control studies are considered. On the other hand, when synthesizing results solely from cohort studies, a reduction in cancer risk was generally observed, but this lacks statistical significance. Similarly, the preventive effect of olive oil on cancer mortality, based on prospective studies, was found to be at the limit of statistical significance [61]. One reason for these marginal findings could be the limited number of cohort studies linking olive oil intake to cancer risk and mortality. Indeed, the most recent meta-analysis by Ke et al. in 2024 included merely seven and six cohort studies regarding tumor incidence and mortality, respectively [61]. Moreover, it is problematic that risk values associated with various organ-specific tumors (breast, skin, lung, bladder, and colon) are aggregated, as these organs likely vary in their sensitivity to olive oil’s preventive effects. Furthermore, most cohort studies only considered olive oils as part of the broader Mediterranean diet, without distinguishing the type of olive oil consumed. The potential cancer-preventive properties of olive oil are supported by numerous in vitro and in vivo studies, highlighting the role of its unique components, such as oleic acid and squalene; however, phenolic compounds, including simple phenols, secoiridoids, and lignans, have garnered the most attention for their substantial bioactive properties [73]. In vitro studies on tumor cells have clearly shown that these compounds can inhibit all stages of the carcinogenesis process (initiation, promotion, and progression) [74]. Animal carcinogenesis model studies have further demonstrated the anti-carcinogenic effects of both olive oil and its phenolic compounds [75]. It is important to note the significant variability in phenolic content across different olive oils [76], ranging from nearly zero in common refined olive oils, to 800 mg/Kg in extra virgin olive oils [76]. Unfortunately, no existing epidemiological studies have considered the phenolic content of the olive oil consumed in association with the risk of cancer. Further prospective studies, precisely calibrated to the type of olive oil consumed, are essential to determine if virgin olive oil may have a cancer-preventive activity.

A recent umbrella review [41] found that a Mediterranean diet and high olive oil consumption significantly reduces stroke risk, but does not significantly affect CHD risk. Moreover, three recent meta-analyses [4,61,67] confirmed the protective role of OO in CVD and stroke and also observed benefits of olive oil for CHD.

Analyzing only high-quality articles (AMSTAR: High, Moderate) that investigated cardiovascular risk factors, such as high blood glucose levels, hypertension, and dyslipidemia, we observed substantial variability in the association of OO and the outcomes. For instance, one study highlighted the beneficial effect of high-phenol OO compared with low-phenol OO on ox-LDL levels and SBP [60], while another found that EVOO intake has a significant impact on insulin level [47]; finally, a third study [48] showed a trivial effect of OO consumption vs. LFD/other oil on serum lipid level and glycemic profile. Results about the importance of the kind of OO used are controversial.

Overweight and obesity are significant risk factors for CVD [77]. Diet composition has a great impact on body composition. The Mediterranean diet, with EVOO as a typical fat source, is often recommended as a strategy to reduce body fat. However, the impact of EVOO on anthropometric indices remains clear. This umbrella review included four meta-analyses on the association between olive oil intake and various anthropometric parameters, including BMI, WC, WC/hip ratio, body fat, weight, and muscle mass, have been considered. The results are often contradictory; only the BMI appears to decrease significantly in olive oil intervention groups compared to controls. This inconsistency may stem from the heterogeneity of the subjects involved (healthy or not), the type of olive oil used (refined, EVOO, whether cooked or raw, in capsules), and the duration of the intervention. Moreover, the specific olive cultivar used can significantly influence the concentration and effects of bioactive molecules [78].

It is well known that chronic inflammation plays a crucial role in the upset of CVDs, which are considered inflammatory diseases [79]. There is substantial evidence of the anti-inflammatory properties of the Mediterranean diet [80]. It is therefore very interesting to evaluate the association between olive oil intake and inflammatory biomarkers. Although the oldest meta-analysis [9] showed a statistically significant reduction in CRP and IL-6 levels in subjects treated with olive oil, more recent meta-analyses [14,47] have not confirmed these findings. However, it should be noted that the older meta-analysis included a much larger sample size (3701) than the more recent ones (485 and 493). In addition, it is possible that, as noted with anthropometric measures, various variables in intervention trials can affect outcomes.

Our results corroborate findings from the study of Toi et al. [40] on the preventive role of OO against type 2 diabetes mellitus (T2DM). The intake of OO appears beneficial for the prevention and management of this disease and it should be promoted at both individual and population levels to mitigate the future burden of T2DM.

Among the outcomes considered in this umbrella review, we found the result of the meta-analysis on pressure ulcers (PUs) of particular interest [71]. Despite including only four RCTs, the analysis indicated that topical application of olive oil reduces the incidence of PUs by both extending the time of development and reducing days of hospitalization. This finding complements the benefits of EVOO on wound healing [81]. Because of its affordability, olive oil can represent a frequently used option for this injury.

### Strengths and Limitations

Umbrella reviews, which analyze the overall results of previously published meta-analyses, represent a valuable tool for defining evidence-based public health choices; however, they may contain some limitations. Specifically, in this case, (i) according to AMSTAR-2, only three publications of the thirty-one selected publications are of high quality, while twenty publications (approximately 65%) are of critically low quality; (ii) studies on the effect of foods or nutrients are challenging to interpret due to variability in the methods used to quantify intake and the presence of different diets followed by the recruited populations; (iii) it must also be considered that the chemical composition of olive oil can vary significantly based on the cultivar, the degree of ripeness of the olives at harvest, and the pressing technique used [78], factors that alter the concentration and type of polyphenols, bioactive compounds crucial to the health effects of olive oil [82].

The results highlighted by this umbrella review underline the need to carry out further intervention trials using a common protocol in terms of time of intervention, type of olive oil used, characteristics of the population, and comparator, especially when results are expressed as MD.

## 5. Conclusions

The evidence synthesized in this umbrella review underscores the multifaceted health benefits associated with olive oil consumption, emphasizing its role in reducing the risk of chronic diseases such as cancer, cardiovascular diseases, and type 2 diabetes. However, significant heterogeneity exists among studies, particularly regarding the types of olive oil used, the methods of administration, and population characteristics. The preventive potential of olive oil against cancer is supported by both in vitro and in vivo studies, highlighting the importance of its phenolic compounds. The evidence suggests a preventive activity in most cancer types, including breast cancer and digestive system cancers.

The health-promoting effects of olive oil result from a plethora of bioactive compounds such as monounsaturated fats, phenolic compounds, polyphenols, and vitamins that, by their anti-inflammatory and antioxidant properties [83], their epigenetic modification ability [84], and their microbiota regulation [85], contribute to the prevention of different chronic degenerative diseases.

However, the absence of epidemiological studies that consider the phenolic concentration of olive oil necessitates further research to ascertain the specific effects of various olive oil types on cancer prevention. The review also highlights the protective role of olive oil in cardiovascular health. Significant associations were found between olive oil consumption and reduced risk of stroke and cardiovascular disease, though results for coronary heart disease remain inconsistent. Notably, the mode of olive oil administration appears to influence health outcomes, with liquid oil showing benefits for blood pressure regulation, unlike capsule forms. Concerning diabetes, our findings confirm that olive oil intake could be advantageous for the prevention and management of type 2 diabetes. This is supported by evidence showing a significant impact of olive oil on glucose homeostasis and insulin sensitivity, suggesting its promotion at both individual and population levels to mitigate the burden of type 2 diabetes. The impact of olive oil on anthropometric indices and inflammatory biomarkers remains inconclusive, likely due to variability in study designs, populations, and olive oil types. While some studies report beneficial effects on BMI and inflammatory markers, others do not, pointing to the need for standardized intervention protocols in future research.

In summary, while olive oil exhibits considerable potential in promoting health and preventing disease, consistent methodological approaches in future trials are essential to validate these findings and clarify the specific conditions under which olive oil can be most beneficial.

## Figures and Tables

**Figure 1 foods-13-02619-f001:**
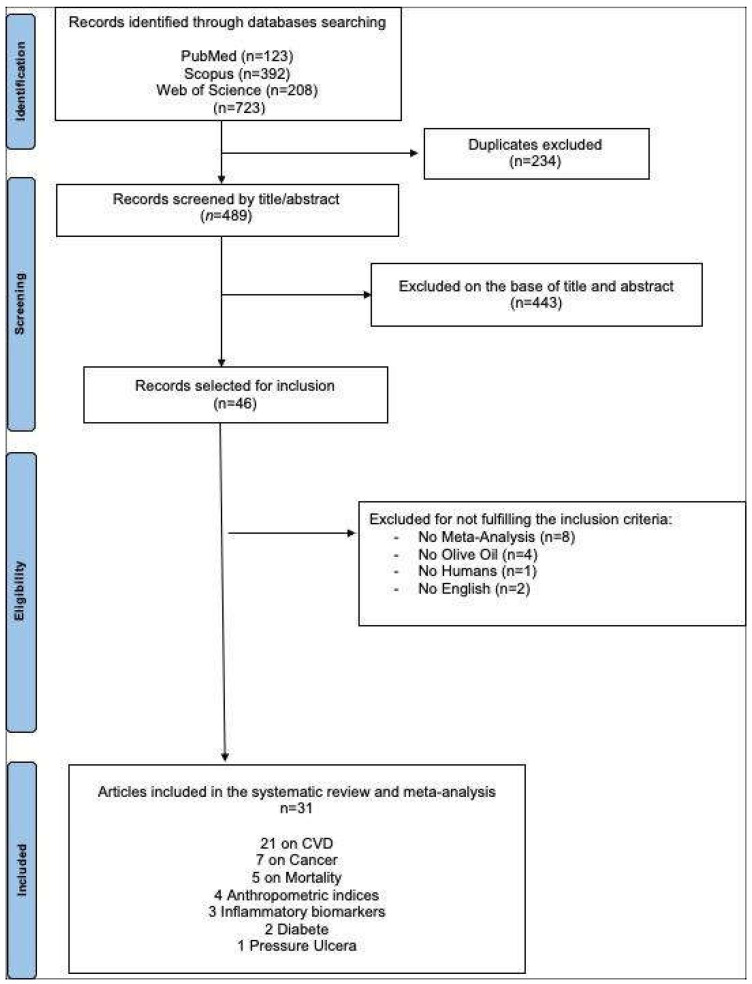
PRISMA flow chart of study selection.

**Table 1 foods-13-02619-t001:** Characteristics of studies that consider outcomes described with risk.

OUTCOME Cancer										
Author, Year (Reference)	Cancer Site	N° Studies Design	Comparison	ES	Study Population	N° Events	N° Total/ Controls	Effect Size (95% CI) Random Model	I^2^ (%)	Quality
Ke, 2024 [61]	All sites	7 CO	Highest vs. lowest	RR	M/F	24,353	1,648,841	1.00 (0.96–1.04)	9.8	Low
	All sites	7 CO	10 g/d increase	RR	M/F	24,353	1,648,841	0.99 (0.97–1.01)	16	
Martínez-Gonzalez, 2022 [4]	Breast	2 CO 1 RCT	25 g/d increase	RR	F	1504	81,243	0.88 (0.65–1.19)	75.6	High
	Other sites	2 CO	25 g/d increase	RR	M/F	884	530,317	0.92 (0.56–1.52)	16.7	
Markellos, 2022 [62]	All sites	6 CO 2 M 37 CC	Highest vs. lowest	RR	M/F	29,830	958,065	0.69 (0.62–0.77)	75.4	Moderate
	All sites	6 CO 2 M	Highest vs. lowest	RR	M/F	12,461	929,771	0.90 (0.77–1.05)	51.7	
	All sites	37 CC	Highest vs. lowest	OR	M/F	17,369	28,294	0.65 (0.57–0.74)	67.2	
	Breast	3 CO 11 CC	Highest vs. lowest	RR	F	N.R	N.R.	0.67 (0.52–0.86)	82.5	
	Breast	3 CO	Highest vs. lowest	RR	F	N.R.	N.R.	0.67 (0.29–1.56)	77.6	
	Breast	11 CC	Highest vs. lowest	RR	F	N.R.	N.R.	0.63 (0.45–0.87)	79.5	
	Gastrointestinal	2 CO 13 CC	Highest vs. lowest	RR	M/F	N.R.	N.R.	0.77 (0.66–0.89)	40.6	
	Colorectal	1 CO 6 CC	Highest vs. lowest	RR	M/F	N.R.	N.R.	0.90 (0.79–1.03)	0.0	
	Upper aerodigestive	6 CC	Highest vs. lowest	RR	M/F	N.R.	N.R.	0.74 (0.60–0.91)	32.7	
	Urinary trac	6 CC	Highest vs. lowest	RR	M/F	N.R.	N.R.	0.46 (0.29–0.72)	72.9	
Sealy, 2021 [63]	Breast	1 CO 1 RCT	Highest vs. lowest	RR	F	1291	68,237	0.48 (0.09–2.79)	89	Low
	Breast	8 CC	Highest vs. lowest	OR	F	5739	13,199	0.76 (0.54–1.06)	82	
Xin, 2015 [64]	Breast	3 CO	Highest vs. lowest	OR	F	N.R.	N.R.	0.89 (0.62–1.29)	76.1	Critic. low
	Breast	9 CC	Highest vs. lowest	OR	F	N.R.	N.R.	0.68 (0.52–0.89)	76.7	
Psaltopoulou, 2011 [65]	All sites	19 CC	Highest vs. lowest	OR	M/F	13,800	23,340	0.66 (0.59–0.75)	62	Critic. low
	Breast	5 CC	Highest vs. lowest	OR	F	N.R.	N.R.	0.64 (0.46–0.89)	N.R.	
	Digestive	8 CC	Highest vs. lowest	OR	M/F	N.R.	N.R.	0.70 (0.61–0.81)	N.R.	
	Others	6 CC	Highest vs. lowest	OR	M/F	N.R.	N.R.	0.66 (0.55–0.79)	N.R.	
Pelucchi, 2011 [66]	Breast	5 CC	Highest vs. lowest	RR	F	N.R.	N.R.	0.62 (0.44–0.88)	N.R.	Critic. low
OUTCOME Cardiovascular										
Author, year (reference)	Type	N° studies design	Comparison	ES	Study Population	N° events	N° total/ controls	Effect size (95% CI) Random model	I^2^ (%)	Quality
Ke, 2024 [61]	CVD	6 CO	Highest vs. lowest	RR	M/F	14,021	168,574	0.85 (0.77–0.93)	40.7	Low
		5 CO	10 g/d increase	RR	M/F	N.R.	N.R.	0.93 (0.88–0.98)	74.1	
	CHD	5 CO	Highest vs. lowest	RR	M/F	8190	170,761	0.85 (0.72–0.99)	60.3	
		4 CO	10 g/d increase	RR	M/F	N.R.	N.R.	0.94 (0.87–1.01)	78.9	
	Stroke	4 CO	Highest vs. lowest	RR	M/F	5045	145,428	0.93 (0.80–1.09)	45.2	
		3 CO	10 g/d increase	RR	M/F	N.R.	N.R.	0.95 (0.92–0.98)	24.3	
Martínez-Gonzalez, 2022 [4]	CVD	7 CO	25 g/d increase	RR	M/F	49,223	806,203	0.83 (0.74–0.94)	50.4	High
	CHD	2 CO	25 g/d increase	RR	M/F	N.R.	N.R.	1.04 (0.83–1.31)	55.6	
	Stroke	2 CO	25 g/d increase	RR	M/F	N.R.	N.R.	0.74 (0.61–0.91)	20.7	
Xia, 2022 [67]	CVD	8 CO	Highest vs. lowest	RR	M/F	14,033	261,016	0.85 (0.77–0.93)	41	Critic. low
		5 CO	5 g/d increase	RR	M/F	N.R.	N.R.	0.96 (0.93–0.99)	67	
Grosso, 2017 [68]	CVD	N.R.	N.R.	RR	M/F	N.R.	N.R.	0.83 (0.77–0.89)	0	Critic. low
Schwingshackl, 2014 [69]	CVD	7 CO	Top vs. bottom third	RR	M/F	N.R.	N.R.	0.72 (0.57–0.91)	75	Critic. low
	CHD	4 CO	Top vs. bottom third	RR	M/F	N.R.	N.R.	0.80 (0.57–1.14)	77	
	Stroke	2 CO	Top vs. bottom third	RR	M/F	N.R.	N.R.	0.60 (0.47–0.77)	0	
Martínez-Gonzalez, 2014 [70]	CVD	3 CC 5 CO 1 RCT	25 g/d increase	RR	M/F	3436	141,860	0.82 (0.70–0.96)	77	Critic. low
	CHD	3 CC	25 g/d increase	RR	M/F	1526	1727	0.73 (0.44–1.21)	89	
	CHD	3 CO 1 RCT	25 g/d increase	RR	M/F	1367	101,460	0.96 (0.78–1.18)	72.5	
	Stroke	2 CO 1 RCT	25 g/d increase	RR	M/F	543	38,673	0.74 (0.60–0.92)	24.4	
OUTCOME Diabetes										
Martínez-Gonzalez, 2022 [4]	Type 2 diabetes	3 CO 1 RCT	25 g/d increase	RR	M/F	13,389	680,239	0.78 (0.69–0.87)	0	High
Schwingshackl, 2017 [7]	Type 2 diabetes	4 CO 1 RCT	Highest vs. lowest	RR	M/F	19,081	N.R.	0.84 (0.77–0.92)	22	High
	Type 2 diabetes	4 CO	10 g/d increase	RR	M/F	18,900	N.R.	0.91 (0.87–0.95)	0	
OUTCOME Pressure Ulcers										
Hernández-Vásquez, 2022 [71]	Pressure ulcers	4 RCT	Olive oil vs. others	RR	M/F	105	1344	0.56 (0.39–0.79)	0	High
	Adverse effects	3 RCT	Olive oil vs. others	RR	M/F	4	1274	0.39 (0.06–2.62)	0	
OUTCOME Mortality										
Author, year (reference)	Causes	N° studies design	Comparison	ES	Study Population	N° events	N° total/ controls	Effect size (95% CI) Random model	I^2^ (%)	Quality
Ke, 2024 [61]	All causes	14 CO	Highest vs. lowest	RR	M/F	176,729	723,224	0.85 (0.81–0.89)	88.4	Low
	CVD	10 CO	Highest vs. lowest	RR	M/F	49,257	702,831	0.77 (0.67–0.80)	78	
	CHD	1 CO	Highest vs. lowest	RR	M/F	226	19,263	1.23 (0.71–2.16)	0	
	Stroke	2 CO	Highest vs. lowest	RR	M/F	5714	540,383	1.04 (0.67–1.62)	43.6	
	Cancer	6 CO	Highest vs. lowest	RR	M/F	56,569	674,834	0.89 (0.79–1.00)	79.1	
Martínez-Gonzalez, 2022 [4]	Cancer	5 CO	25 g/d increase	RR	M/F	56,487	673,502	0.94 (0.85–1.05)	62.8	High
	All causes	10 CO 1 RCT	25 g/d increase	RR	M/F	174,081	733,420	0.89 (0.85–0.93)	65.2	
Xia, 2022 [67]	All causes	11 CO	Highest vs. lowest	RR	M/F	173,817	713,000	0.83 (0.77–0.90)	93	Critic. low
	All causes	5 CO	5 g/d increase	RR	M/F	N.R.	N.R.	0.96 (0.95–0.96)	0	
Eleftheriou, 2018 [72]	All cause	4 CO	Above vs. below median	RR	M/F	N.R.	N.R.	0.97 (0.82–1.15)	N.R.	Critic. low
Schwingshackl, 2014 [69]	All causes	5 CO	Top vs. bottom third	RR	M/F	N.R.	84,988	0.77 (0.71–0.84)	0	Critic. low
	CVD	5 CO	Top vs. bottom third	RR	M/F	N.R.	N.R.	0.70 (0.48–1.03)	71	
	Stroke	2 CO	Top vs. bottom third	RR	M/F	N.R.	N.R.	0.60 (0.47–0.67)	0	
	CHD	4 CO	Top vs. bottom third	RR	M/F	N.R.	N.R.	0.80 (0.57–1.14)	77	

**Table 2 foods-13-02619-t002:** Characteristics of studies that consider outcomes described with MD.

Author, Year (Reference)	Population Characteristics	Oil Type vs. Comparison	N° Studies (Design: RCTs)	Outcome	Effect Size: MD, SMD, WMD (95% CI) Effect Model: Random/Fixed	I^2^ (%)	Quality
Tsamos, 2024 [48]	Age ≥ 18 y, N° 515 Men and women, healthy and not	Olive oil vs. ^1^ LFD; Other vegetable oils	4	TC	MD = 2.40 (−6.89, 11.70); REM	38	Moderate
			5	HDL	MD = 1.42 (−3.45, 6.29); REM	94	
			4	LDL	MD = 4.77 (−3.19, 12.73); REM	42	
			6	TG	MD = 13.03 (−13.81, 39.87); REM	86	
			6	Glucose	MD = −0.69 (−4.84, 3.46); REM	54	
			5	Insulin level	MD = −0.42 (−3.60, 2.76); REM	95	
Morvaridzadeh, 2024 [47]	Age ≥ 18, N° 2020, men and women, healthy and not	EVOO vs. Regular diet; Other oils; Butter	26	TG	SMD: −0.05; (−0.17, 0.07); REM	19.07	Moderate
			27	TC	SMD: 0.07; (−0.12, 0.26); REM	68.87	
			29	LDL	SMD: 0.05; (−0.12, 0.22); REM	61.09	
			28	HDL	SMD: 0.13; (−0.03, 0.28); REM	54.42	
			6	VLDL	SMD: 0.12; (−0.14, 0.38); REM	30.31	
			10	ApoA-I	SMD: 0.16; (−0.17, 0.50); REM	73.76	
			9	ApoB	SMD: 0.29; (−0.06, 0.63); REM	72.12	
			4	Lp-A	SMD: −0.35; (−1.02, 0.32); REM	83.96	
			16	FBS	SMD: 0.05; (−0.08, 0.18); REM	0	
			10	insulin level	SMD: −0.28; (−0.51, −0.05); REM	48.57	
			9	HOMA-IR	SMD: −0.19; (−0.35, 0.03); REM	0	
			9	SBP	SMD: −0.04; (−0.33, 0.25); REM	63.02	
			9	DBP	SMD: −0.11; (−0.38, 0.16); REM	56.26	
Zupo, 2023 [50]	Age ≥ 18, N° 415, men and women, healthy and not	Olive oil vs. Exposure level (low, medium, high)	8	TC (overall)	MD: 0.49; (−0.55, 1.53); n/a	14	Critically low
			6	TC (low)	MD: 1.42; (−0.34, 3.18); n/a	0	
			5	TC (medium)	MD: −0.05; (−1.73, 1.62); n/a	0	
			7	TC (high)	MD: 0.05; (−1.98, 2.08); n/a	16	
			10	LDL (overall)	MD: −0.83; (−1.67, 0.01); n/a	73	
			6	LDL (low)	MD: 0.81; (−0.86, 2.47); n/a	19	
			6	LDL (medium)	MD: 0.66; (−0.61, 1.92); n/a	0	
			7	LDL (high)	MD: −4.28; (−5.78, −2.77); n/a	83	
			10	HDL	MD: 1.03; (0.68, 1.38); n/a	38	
			6	HDL (low)	MD: 0.66; (0.10, 1.23); n/a	25	
			6	HDL (medium)	MD: 1.36; (0.76, 1.95); n/a	0	
			7	HDL (high)	MD: 1.13; (0.45, 1.80); n/a	70	
Derakhshandeh-Rishehri, 2023 [51]	Age ≥ 18, N° 3062, men and women, healthy and not	^2^ HPOO vs. ^3^ LPOO	5	ox LDL	WMD: −0.29 U/L; (−0.51, −0.07); REM	24.9	Critically low
			5	MDA	WMD: −1.82 μmoL/L; (−3.13, −0.50); REM	94.9	
			3	^4^ FRAP	WMD: 0.0 mmoL/L; (−0.03, 0.04); REM	0	
Fakhri, 2022 [52]	N° 79, men and women, healthy and not	Olive oil vs. Natural products or placebo	3	SBP	SMD: −0.46; (−0.97, 0.04); REM	59.2	Critically low
			3	DBP	SMD: −0.34; (−0.66, −0.03); REM	0	
Jabbarzadeh-Ganjeh B, 2023 [53]	Age ≥ 18 y, N° 6482, healthy and not	EVOO, OO, refined OO, Virgin OO vs. Usual diet; Different oils	31	TC	MD: 0.79 mg/dL; (−0.08, 1.66); REM	57	Critically low
			31	LDL	MD: 0.04 md/dL; (−1.01, 0.94); REM	80	
			34	HDL	MD: 0.22 mg/dL; (−0.01, 0.45); REM	38	
			32	TG	MD: 0.39 mg/dL; (−0.33, 1.11); REM	7	
Pastor, 2021 [54]	Age ≥ 18 y, N° 10,996 Men and women, healthy and not	Olive oil vs. Other vegetable and animal oils; Standard healthy diet; Hypocaloric diet; Diet + nuts or PUFAs; LFD	15	Blood pressure	SMD = −0.00; (−0.06, 0.05); REM	37	Critically low
			n/a	Metabolic syndrome	SMD = −0.01, (−0.05, 0.03); REM	55	
			12	Glycemic profile	SMD = 0.01 (−0.05, 0.06); REM	32	
			n/a	Lipid profile	SMD = 0.01 (−0.05, 0.06); REM	32	
Dehghani, 2021 [55]	Age ≥ 18 y, N° 633, men and women, healthy and not	EVOO vs. Other type of oil	13	FBS	SMD: −0.07; (−0.20, 0.07); REM	0	Critically low
			4	HOMA-IR	SMD: −0.32; (−0.75, 0.10); REM	51.0	
			4	insulin level	SMD: −0.32; (−0.70, 0.06); REM	38.0	
George, 2019 [56]	Age ≥ 18 y, N° 2652 Men and women, healthy and not	HPOO vs. LPOO	3	MDA	MD = −0.07 (−0.12, −0.02); Random	88	Critically low
			5	oxLDL	SMD = −0.44 (−0.78, −0.10); REM	49	
			8	TC	MD = −4.47 (−6.54, −2.39); REM	0	
			8	LDL	MD = −3.54 (−7.27, 0.19); REM	67	
			8	HDL	MD = 2.73 (0.41, 5.04); REM	65	
Tsartsou, 2019 [8] Network meta-analysis	Age ≥ 18 y, N° 7688 Men and women, healthy and not	Olive oil vs. HPOO	6	Glucose	SMD = −0.08 (−0.23, 0.06) ^5^ NR	0	Critically low
			10	TG	SMD = 0.04 (−0.07, 0.15) NR	87	
			12	TC	SMD = 0.05 (−0.05, 0.15) NR	65	
			11	HDL	SMD = −0.16 (−0.26, −0.05) NR	66	
			10	LDL	SMD = 0.09 (−0.03, 0.20) NR	11	
			5	oxLDL	SMD = 0.09 (−0.03, 0.22) NR	0	
		Olive oil vs. LPOO	6	Glucose	SMD = −0.04 (−0.27, 0.18) NR	0	
			6	TG	SMD = 0.01 (−0.12, 0.14) NR	0	
			9	TC	SMD = 0.14 (0.02, 0.25) NR	49	
			8	HDL	SMD = −0.13 (−0.25, 0.00) NR	33	
			8	LDL	SMD = 0.19 (0.06, 0.31) NR	60	
			3	oxLDL	SMD = 0.00 (−0.17, 0.17) NR	0	
Schwingshackl, 2019 [57] Network meta-analysis	Age ≥ 18 y, N° 611 Men and women, healthy and not	^6^ ROO vs. ^7^ MOO	NR	TC	MD = −0.03 (−0.13, 0.08) REM	NR	Critically low
			NR	HDL	MD = −0.00 (−0.03, 0.03) REM	NR	
			2	LDL	MD = −0.06 (−0.24, 0.12) REM	NR	
			NR	oxLDL	SMD = −0.15 (−1.10, 0.79) REM	NR	
			NR	TG	MD = 0.00 (−0.14, 0.15) REM	NR	
		ROO vs. ^8^ LP(E)VOO	1	SBP	MD = −2.87 (−5.39, −0.35) REM	NR	
			NR	TC	MD = −0.00 (−0.18, 0.17) REM	NR	
			NR	HDL	MD = 0.01 (−0.04, 0.05) REM	NR	
			3	LDL	MD = 0.05 (−0.10, 0.20) REM	NR	
			2	oxLDL	SMD = −0.26 (−1.15, 0.73) REM	NR	
			NR	DBP	MD = −0.02 (−3.95, 3.91) REM	NR	
			NR	TG	MD = −0.03 (−0.19, 0.13) REM	NR	
		ROO vs. ^9^ HP(E)VOO	3	SBP	MD = −2.99 (−6.12, −0.15) REM	NR	
			NR	TC	MD = 0.01 (−0.09, 0.10) REM	NR	
			NR	HDL	MD = 0.01 (−0.01, 0.04) REM	NR	
			4	LDL	MD = −0.09 (−0.24, 0.06) REM	NR	
			5	oxLDL	SMD = −0.68 (−1.31, 0.04) REM	NR	
			NR	DBP	MD = −0.10 (−3.05, 3.24) REM	NR	
			NR	TG	MD = 0.02 (−0.11, 0.16) REM	NR	
Ghobadi 2019 [59]	Age ≥ 18, N° 1089, men and women, healthy and not	Olive oil vs. Other type of oil	26	TC	OO: WMD = 6.72, (2.8, 10.6); REM	46.4	Critically low
			12		Virgin Oil: WMD = 6.36, (−1.16, 13.9); REM	70.4	
			6		Refine Oil: WMD = 5.21, (0.72, 9.7); REM	10.4	
			8		Not stated Oil: WMD = 7.7, (0.84, 14.6); REM	0	
			11		ω3 RO: WMD = 6.4, (2, 10.87); REM	19.4	
			12		ω6 RO. WMD = 9.9, (2.75, 17); REM	52.1	
			3		SFA Rich oil: WMD = 2.2, (−9.04, 13.4); REM	60	
			5		MO: WMD = 6.5, (1.11, 11.8); REM	0	
			24	LDL	OO, WMD = 4.2, (1.4, 7.01); REM	23	
			12		Virgin Oil: WMD = 3.36, (−1.33, 8.05); REM	34	
			6		Refine Oil: WMD = 5.04, (−0.96, 11.1); REM	57.3	
			7		Not stated Oil: WMD = 4.84, (−1.32, 11); REM	0	
			12		ω3 RO: WMD = 3.3, (−0.2, 6.7); REM	4.2	
			10		ω6 RO: WMD = 5.23, (−0.2, 10.5); REM	22.6	
			3		SFA Rich oil: WMD = 3.8, (−5.78, 13.4); REM	50.3	
			5		MO: WMD = 6.43, (2, 11); REM	0	
			26	HDL	OO: WMD = 1.37, (0.4, 2.36); REM	0	
			12		Virgin Oil: WMD = 1.02, (−0.6, 2.64); REM	4.4	
			5		Refine Oil: WMD = 1.02, (−0.9, 2.9); REM	6.6	
			9		Not stated Oil: WMD = 2.3, (0.4, 4.2); REM	0	
			12		ω3 RO: WMD = 1.9, (0.5, 3.25); REM	0	
			11		ω6 RO: WMD = 0.75, (−1.4, 2.9); REM	0	
			3		SFA Rich oil: WMD = 0.76, (−1.2, 2.7); REM	0	
			5		MO: WMD = 4.11, (0.95, 7.3); REM	49.1	
			25	TG	OO: WMD = 4.31, (0.5, 8.12); REM	0	
			12		Virgin Oil: WMD = 7.75, (2.77, 12.7); REM	0	
			4		Refine Oil: WMD = −1.22, (−17.8, 15.4); REM	24.2	
			9		Not stated Oil: WMD = 3.32, (−4.4, 11); REM	0	
			11		ω3 RO: WMD = 8.32, (2.66, 13.9); REM	0	
			10		ω6 RO: WMD = 5.51, (−1.2, 12.2); REM	0	
			3		SFA Rich oil: WMD = −8.3, (−17.4, 0.8); REM	0	
			5		MO: WMD = 3.9, (7.7, 15.6); REM	0	
			10	Apo A1	OO, WMD = 4.3, (−0.43, 9.01); REM	36.4	
			10	Apo B	OO, WMD = 4.05, (−0.64, 8.75); REM	38.6	
Zamora-Zamora, 2018 [58]	Age ≥ 18, N° 6651, men and women, healthy and not	Olive oil vs. Diet; Usual diet; LFD; Other type of oil	15	DBP	OO: MD = −0.73, (−1.07, −0.40); FEM	84.5	Critically low
					Capsules OO: MD = 0.14, (−0.35, 0.64); FEM	76.2	
					Liquid OO: MD = −1.44, (−1.89, −1); FEM	86.9	
			13	SBP	OO: MD = −0.11, (−0.68, 0.46); FEM	85.1	
					Capsules OO: MD = 0.17, (−0.72, 1.06); FEM	87.5	
					Liquid OO: MD = −0.31, (−1.06, −0.44); FEM	83.2	
Schwingshackl, 2017 [7]	Age ≥ 18, N° 3152, men and women, healthy and not	Olive oil vs. other oil; Lowest OO intake	22	HbA1c	MD = −0.27, (−0.37, −0.17); REM	0	high
			25	FBS	MD = −0.44, (−0.66, −0.22); REM	26	
Hohmann, 2015 [60]	Age ≥ 18, N° 417, men and women, healthy and not	Olive oil (HPOO) vs. LPOO	2	SBP	SMD = −0.52, (−0.77, −0.27); REM	32	Moderate
			2	DBP	SMD = −0.20, (−1.01, 0.62); REM	94	
			6	LDL	SMD = −0.03, (−0.15, 0.09); REM	39	
			4	oxLDL	SMD = −0.25, (−0.50, −0.00); REM	80	
			6	TC	SMD = −0.05, (−0.16, 0.05); REM	0	
			6	HDL	SMD = −0.03, (−0.14, 0.08); REM	34	
			5	TG	SMD = 0.02, (−0.22, 0.25); REM	91.9	
			2	MDA	SMD = −0.02, (−0.20, 0.15); REM	40	
OUTCOME: Anthropometric indices							
Tsamos, 2024 [48]	Age ≥ 18 y; N° 1502, men and women, BMI: 32.45 kg/m^2^ at baseline; unhealthy subjects	Olive oil vs. LFD; Other vegetable oils	7	BMI	MWD: −0.57 (−1.08, −0.06); REM	51	Moderate
			6	WC	MWD: −0.23 (−1.23, 0.76); REM	0	
Morvaridzadeh, 2024 [47]	Age ≥ 18, N° 1372, men and women, healthy and not	EVOO vs. Regular diet; Other vegetable oils; Animal fats	17	BMI	SMD: −0.04 (−0.17, 0.09); REM	0	Moderate
			13	WC	SMD: −0.01 (−0.16, 0.13); REM	0	
			3	WC/Hip	SMD: −0.08 (−0.44, 0.27); REM	0	
			16	Weight	SMD: −0.06 (−0.19, 0.07); REM	0	
Santos, 2023 [49]	Age ≥ 18, N° 1208, men and women	EVOO or Virgin Olive oil or Refined olive oil vs. Other vegetable and animal oils; Standard healthy diet; Hypocaloric diet; Diet + nuts or PUFAs LFD	41	BMI	MD: −0.05 (−0.23, 0.13); FEM	0	Moderate
			33	WC	MD: 0.28 (−0.22, 0.78); FEM	15	
			9	Hip C	MD: 1.31 (−0.24, 2.86); FEM	0	
			12	WC/Hip	MD: 0.01 (0.0, 0.02); FEM	0	
			18	Total Body fat (kg)	MD: 0.24 (−0.85, 0.37); REM	33	
			22	Total Body fat (%)	MD: 0.02 (−0.57, 0.61); FEM	0	
			20	Muscle mass	MD: −0.27 (−0.58, 0.05); FEM	11	
Pastor, 2021 [54]	Age ≥ 18 y; N° 1912, men and women, healthy and not	EVOO or OO vs. Other vegetable and animal oils; Standard healthy diet; Hypocaloric diet; Diet + nuts or PUFAs; LFD	17	Body composition	SMD: −0.02 (−0.10, 0.05); REM	18	Critically low
OUTCOME: Inflammatory biomarkers							
Morvaridzadeh, 2024 [47]	Age ≥18 y; N° 493 men and women, healthy and not	EVOO or OO vs. Regular diet; Other oils; Butter	7	CRP	SMD: 0.03 (−0.20, 0.26); REM	21	Moderate
			5	IL-6	SMD: −0.07 (−0.16, 0.30); REM	0	
			3	IL-10	SMD: −0.06 (−0.33, 0.21); REM	0	
			3	TNF-a	SMD: 0.03 (−0.32, 0.38); REM	0	
Fernandes M. Sc., 2020 [14]	Age >18 y; N° 485 women and men; healthy and not	EVOO or OO vs. ^10^ SFA-rich diet; LFD; Other oil	4	CRP	MWD: −0.30 (−1.46, 0.86); REM	92	Moderate
			3	IL-6	MWD: −0.60 (−0.64, −0.56); REM	0	
Schwingshackl, 2015 [9]	Age ≥19 y; N° 3701, women and men; healthy and not	EVOO or OO liquid or capsules vs. LFD; Diet + nuts or PUFAs; Other vegetable and animals oils; Healthy diet	14	CRP	MD: −0.64 (−0.96, −0.31); REM	66	Critically low
			7	IL-6	MD: −0.29 (0.07, −0.02); REM	62	
			8	Flow mediated dilatation	MD: 0.76 (0.27, 1.24); REM	26	
			6	Adiponectin	MD: 0.44 (0.20, 1.09); REM	56	
			5	TNF-a	MD: 0.02 (0.02, 0.07); REM	95	
			2	sE-Selectin	MD: 3.16 (4.07, 2.25); REM	0	
			4	sP-Selectin	MD: 10.78 (4.01, 17.54); REM	41	
			7	sICAM-1	MD: 0.02 (0.04, 0.00); REM	84	
			8	sVCAM-1	MD: 0.02 (0.05, 0.01); REM	37	

^1^ low-fat diet; ^2^ high-phenol olive oil; ^3^ low-phenol olive oil; ^4^ ferric-reducing ability of plasma; ^5^ not reported; ^6^ refined olive oil; ^7^ mixed olive oil; ^8^ low-phenolic (extra) virgin olive oil; ^9^ high-phenolic (extra) virgin olive oil; ^10^ saturated fatty acid-rich diet.

## Data Availability

No new data were created or analyzed in this study. Data sharing is not applicable to this article.

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
