# Peer review of "Health Outcomes Associated with Olive Oil Intake: An Umbrella Review of Meta-Analyses"

_foods, 2024, doi:10.3390/foods13162619_

Round 1

Reviewer 1 Report

Comments and Suggestions for Authors

Journal: MDPI_foods 

Title: Health Outcomes Associated with Olive Oil Intake: An Umbrella Review of Meta-Analyses 

Manuscript ID: foods_3126407 

Comments

This manuscript reviewed the “Health Outcomes Associated with Olive Oil Intake: An Umbrella Review of Meta-Analyses”. Studies indicate that olive oil consumption is associated with reduced risks of various chronic diseases. A systematic review was conducted to evaluate the overall health outcomes linked to olive oil intake. From 723 identified articles, 24 were included in the review. Previous efforts to systematically appraise the evidence on olive oil have focused on single disease endpoints (e.g., CVD). Instead, in this review authors used the umbrella review methodology (i.e., the syntheses of existing systematic reviews with meta-analyses) to capture the full spectrum of outcomes associated with olive oil intake.

Unlike previous papers, this paper is considered to be able to provide useful information by reviewing the correlation between olive oil intake and various diseases.

However, to submit this paper to the MDPI foods Journal, the minor comments need to be revised as followings.

1.      The explanations (i.e. comparison between outcome and effect size) and discussions related to the indicators presented in Table 1 and Table 2 that summarize the literature are insufficient.

Author Response

Comment: Unlike previous papers, this paper is considered to be able to provide useful information by reviewing the correlation between olive oil intake and various diseases. However, to submit this paper to the MDPI foods Journal, the minor comments need to be revised as followings.

The explanations (i.e. comparison between outcome and effect size) and discussions related to the indicators presented in Table 1 and Table 2 that summarize the literature are insufficient.

ANSWER: Dear reviewer, we are not sure to have clearly understood your point. In particular could you explain what “comparison between outcome and effect size” means? Secondly if we address with “indicators” the outcomes, I understand that a deeper explanation of outcomes’ meaning could help to read the umbrella review’s results, but considering the number of outcomes and the target of readers to who Foods refers, we think this level of explanation is enough.

Reviewer 2 Report

Comments and Suggestions for Authors

The topic of this manuscript fits well the scope of FOODS. However, as it is well known that olive oil displays many health-promoting effects, this article only possesses limited scientific values. As FOODS is an OA journal, the authors should make some minor amendments.

(1) Why the authors include all articles into their study? They should only focus on the not that old literature (published in recent certain years). 

(2) Olive oil has many different grade? Did the authors consider this factor? 

(3) Is race / genetic background has an association with the health-promoting effects of olive oil?

(4) Did the authors also detect any adverse effects of olive oil reported literature? There should be a balanced interpretation of the literature.

(5) The authors should briefly discuss the limitation of their study.

(6) The authors should briefly discuss the potential mechanism of action of the health-promoting effects of olive oil.

Author Response

(1) Comment: Why the authors include all articles into their study? They should only focus on the not that old literature (published in recent certain years). 

ANSWER: Thank you for your comment.  We have chosen to include also older publications to give a comprehensive overview about the topic, actually we focused on the recent literature in discussion.

(2) Comment: Olive oil has many different grade? Did the authors consider this factor? 

ANSWER: Thank you for your comment, this is a very interesting topic. Unfortunately meta-analyses included did not always uniquely define the grade of olive oil used, so it is not possible  to address this aspect. For this reason, this aspect was improved in the limitations paragraph.

(3) Comment: Is race / genetic background has an association with the health-promoting effects of olive oil?

ANSWER: Meta-analyses included did not considered the race/genetic background, so it was not possible study the association.

(4) Comment: Did the authors also detect any adverse effects of olive oil reported literature? There should be a balanced interpretation of the literature.

ANSWER: Thank you for your comment. Certainly, for a balanced interpretation of the literature, any adverse effects associated with olive oil consumption should also be considered but the meta-analyses included did not mention any adverse effects, so we could not report them.

(5) Comment: The authors should briefly discuss the limitation of their study.

ANSWER: Thank you for your comment, we have modified the discussion paragraph as follows:

At Lines 427-439 you will find:

“4.1 Strengths and limitations.

Umbrella reviews, which analyze the overall results of previously published meta-analyses, represent a very interesting tool for defining evidence-based public health choices, however they may contain some limitations. In particular, in this case: i) according to AMSTAR-2 only 3 publications of the 31 selected are of high quality, 20 publications (about 65%) are of critically low quality; ii) studies on the effect of foods or nutrients are difficult to interpret due to both the variability in the methods of quantifying intake and the presence of different diets followed by the recruited populations; iii) it must be considered that the chemical composition of olive oil can differ greatly depending on the cultivar, the degree of ripeness of the olives at the time of harvesting and the pressing technique (85: Fabiani, R., Sepporta, M.V., Mazzam T., Rosignoli, P., Fuccelli, R., De Bartolomeo, A., Crescimanno, M., Taticchi, A., Esposto, S., Servili, M., Morozzi, G. Influence of cultivar and concentration of selected phenolic constituents on the in vitro chemiopreventive potential of olive oil extracts. J. Agric. Food Chem. 2011; 59(15): 8167-74. doi: 10.1021/jf201459u.), all that modifies the concentration and type of polyphenols, bioactive compounds involved in the health effects of olive oil (89: Polyphenols, the Healthy Brand of Olive Oil: Insights and Perspectives. Finicelli M, Squillaro T, Galderisi U, Peluso G. Nutrients. 2021 Oct 27;13(11):3831. doi: 10.3390/nu13113831. PMID: 34836087).”

(6) Comment: The authors should briefly discuss the potential mechanism of action of the health-promoting effects of olive oil.

ANSWER: Thank you for your comment. We have improved the conclusion paragraph as follows:

From Line 455 to line 459: “The health-promoting effects of olive oil result from a plethora of bioactive compounds such as monounsaturated fats, phenolic compounds, polyphenols, vitamins, that, by their anti-inflammatory and antioxidant properties (90:Olive Polyphenols: Antioxidant and Anti-Inflammatory Properties. Bucciantini M, Leri M, Nardiello P, Casamenti F, Stefani M. Antioxidants (Basel). 2021 Jun 29;10(7):1044.)  their epigenetic modifications ability (91 Epigenetic Modifications Induced by Olive Oil and Its Phenolic Compounds: A Systematic Review. Fabiani R, Vella N, Rosignoli P. Molecules. 2021 Jan 7;26(2):273.) and their microbiota regulation (92: Impact of fundamental components of the Mediterranean diet on the microbiota composition in blood pressure regulation. Zambrano AK, Cadena-Ullauri S, Ruiz-Pozo VA, Tamayo-Trujillo R, Paz-Cruz E, Guevara-Ramírez P, Frias-Toral E, Simancas-Racines D. J Transl Med. 2024 May 3;22(1):417.) contribute to the prevention of different chronic degenerative diseases.”

Reviewer 3 Report

Comments and Suggestions for Authors

The authors' review focuses on the health effects of olive oil. This review is well written overall. However, some small details need to be revised. Therefore, I suggest to do minor revision.

The specific content is as follows:

1. Abstract: Background, Objective, Methods, Results, please check whether these four words should be deleted according to the journal format requirements.

2. There are too many keywords. Five to six are recommended.

3. Lines 45-48, “……a higher intake of olive oil is associated with a reduced risk of several chronic diseases” was mentioned by the authors. It is recommended that the authors follow this sentence with some references to elaborate on the fact that olive oil reduces the risk of various diseases.

4. 2. Materials and Methods, 3. Results, 4. Discussion: The author's article type is a review paper, and these titles does not seem to be appropriate.

5. Supplementary Table S1 is missing.

Author Response

  1. Abstract: Background, Objective, Methods, Results, please check whether these four words should be deleted according to the journal format requirements.

ANSWER: According to the journal format requirements we modified the abstract in a single paragraph, following the style of structured abstracts, but without headings.

  1. There are too many keywords. Five to six are recommended.

ANSWER: We have removed exceeding keywords. Now there are five keywords, as recommended.

  1. Lines 45-48, “……a higher intake of olive oil is associated with a reduced risk of several chronic diseases” was mentioned by the authors. It is recommended that the authors follow this sentence with some references to elaborate on the fact that olive oil reduces the risk of various diseases.

ANSWER: As you suggested, we introduced (lines 44-49) references to provide a background to the positive effects on cardiovascular health, diabetes management, blood pressure, inflammation, and metabolic functions underscore the importance of incorporating olive oil into a balanced diet for long-term health benefits.

  1. Materials and Methods, 3. Results, 4. Discussion: The author's article type is a review paper, and these titles does not seem to be appropriate.

ANSWER: Thank you for your suggestion, but we have written the manuscript according to the journal format "Structured reviews and meta-analyses should use the same structure as research articles and should ensure they conform to the PRISMA guidelines".

  1. Supplementary Table S1 is missing.

ANSWER: We apologize for the oversight. We have attached Supplementary Table S1